# The Seamless Communication on a Rural Island in Japan: A Qualitative Study from the Perspective of Healthcare Professionals

**DOI:** 10.3390/ijerph18189479

**Published:** 2021-09-08

**Authors:** Moe Kuroda, Ryuichi Ohta, Kaku Kuroda, Seiji Yamashiro, Keiichiro Kita

**Affiliations:** 1Department of General Medicine, Toyama University Hospital, Toyama 930-0194, Japan; kaku555552007@gmail.com (K.K.); yamashir@med.u-toyama.ac.jp (S.Y.); keikita@med.u-toyama.ac.jp (K.K.); 2Department of Community Care Medicine, Unnan City Hospital, Unnan 699-1221, Japan; ryuichiohta0120@gmail.com; 3Department of Family Medicine, SUNY Upstate Medical University, Syracuse, NY 13215, USA; 4Department of Primary Care, Toyama University Hospital, Toyama 930-0194, Japan

**Keywords:** interprofessional collaboration, seamless communication, informal resources, flexibility, rural health

## Abstract

On remote islands, interprofessional collaboration is essential to support older adults who live at home, despite the limited number of healthcare professionals (HCPs). Therefore, it is important for HCPs to collect and share information about older adults with health problems. This study aimed to clarify how rural HCPs collaborate using limited resources to support older adults in remote islands. We conducted semi-structured interviews with 10 healthcare providers for older adults on Zamami Island of Okinawa, Japan. We performed a qualitative analysis using the steps for coding and theorization method. Four themes were extracted: “Collection and communication of information between residents”, “Communication of information from non-HCPs to HCPs”, “Sharing of information between HCPs”, and “HCPs taking action to initiate their approach”. Islanders take care of each other and know each other’s health status, while HCPs gather their health information. When necessary, HCPs on the island gain essential information regarding older adult patients from islanders not only through work, but also through personal interactions. Afterward, HCPs approach older adults who need health care. The human connections on this remote island serve as social capital and enable flexibility in both gathering information and seamless communication among islanders who also serve as informal resources that contribute support for older adults.

## 1. Introduction

In an aging society, home care is essential to prolong the lives of older adults who desire to live at home or where they are used to [1]. Moreover, interprofessional collaboration is essential in maintaining the sustainability of home care [2]. It enhances both the patients’ experiences and the cost of performing the practice [3]. Furthermore, it is crucial in caring for older adult patients with multiple comorbidities as it enables professionals to manage individuals in a flexible manner [4]. To improve interprofessional collaboration, it is important to consider whether the setting is rural or urban because factors, such as differences in population, location, and accessibility to medical resources, significantly influence the quality of interprofessional collaboration [5].

Interprofessional collaboration is especially essential in rural areas because more effective and efficient health care are necessary in remote locations, for populations with a higher ratio of older adults, and in places with limited medical and human resources [6], where older adults cannot easily access medical facilities [7]. This is provided comprehensively by healthcare professionals (HCPs), which include doctors and nurses who manage the medical aspect, care managers and homecare workers who support patient welfare, and public health nurses and local officials who manage social services and local resources. This is the theoretical framework that outlines the role of HCPs in interprofessional collaboration.

However, it is possible that this theoretical framework cannot be sustained on remote islands where human and medical resources are extremely limited, due to the lack of workforce and aging society [5,8,9,10]. Thus, collaboration should not involve only HCPs, but also informal or nonprofessional resources such as friends, neighborhoods, and local volunteers [11]. Owing to their informal support, older adults residing in rural places can continue to live there.

Okinawa, Japan is made up of remote islands, and 16 islands each have their own prefectural clinic, which is managed by a primary care physician and a nurse [12]. They collaborate with local HCPs such as local public health nurses, care managers, care workers, and other local officials. Since these remote islands are isolated from the main island, helicopter transportation is necessary for patients who require emergency admission [13]. However, weather conditions and nightfall prevent smooth emergency transportation. Thus, regular health management and early disease detection are important in the prevention of emergency admissions [14,15]. This is especially difficult for older adults who live alone in a remote island since they cannot easily detect minor changes and they try to hide signs and symptoms of disease so as not to bother others [16]. HCPs play a significant role in addressing those health vulnerabilities. However, there aren’t enough HCPs, so they work together with residents of the island to provide support.

Studies have demonstrated the process of collecting information and communication between non-HCPs and HCPs in rural areas [17,18]. In terms of interprofessional collaboration in remote islands, the importance of collaboration among HCPs for supporting home care of older adults has been previously pointed out [19], as well as the importance of mutual supports between islanders [20]. However, there is scant evidence on how non-HCPs and HCPs communicate with each other in the context of remote islands. This study aimed to clarify how rural HCPs collaborate with each other using limited resources to support older adults in remote islands via interviews with HCPs who work in a remote island in Okinawa.

## 2. Materials and Methods

### 2.1. Setting

Zamami Island is a small and remote island with a population of 603 people (as of April 2018) and consists of three districts. It is located approximately 40 km northwest of the main island of Okinawa as shown in Figure 1, and public transportation to the main island consists of high-speed boats (50 min, one way) and ferries (two hours, one way) [21]. The only available medical institution is a prefectural clinic consisting of a solo doctor and a nurse. Healthcare services are provided in collaboration with the local government (including public health nurses), the Council of Social Welfare, and daycare and short-stay service facilities. If emergency hospitalization is needed, patients are transported to the mainland via emergency airlift, using a medical helicopter, or with the help of self-defense forces. However, weather and transportation issues occasionally interrupt the smooth transfer, and these situations can result in the worsening of the patient’s status. Therefore, regular health management and the early detection of diseases are necessary.

### 2.2. Participants

Participants included HCPs who worked for older adults in Zamami Island between April 2017 and March 2018. Besides the physician who was the author of this study, HCPs included nurses, public health nurses, care managers, care workers, the staff of the council of social and welfare, and the local office staff in charge of caring for older adults. A total of 10 HCPs who were selected via purposive sampling were interviewed, including one nurse, two public health nurses, two staff members of the council of social and welfare, one local office staff member in charge of caring for older adults, three care managers, and one care worker. These 10 participants covered all HCPs in this island except for care workers and a physician. The mean age of participants was 47.4 years (SD 11.66, range 25–70), and the mean duration of interviews was 43 min. Among the participants, 80% were women and 20% were men.

At the time of the interview, participants had resided on the island for more than one year. Among them, five participants had lived on the island for more than 20 years, including residents of the island; these participants were familiar with the residents. To protect their privacy, details regarding their age and profession were kept confidential.

### 2.3. Data Collection Method

The participants were individually invited to participate in a semi-structured interview on different days. These one-on-one interviews were conducted in a private room for 30 min to 1 h. We used one-on-one semi-structured interviews, which supported the participants in reflecting and concentrating on their personal experiences and verbalizing their feelings without hesitation [18].

We used an interview guide to conduct the interviews depending on the specific case. The interview required participants to recall an experience wherein interprofessional collaboration led to the successful early detection of disease and approach in a patient before their status became severe. The participants were then asked the following questions: “How do you acquire information related to older adults on the island?”, “What sources of information are available?” “How do you share the information obtained?”, and “What is needed to prevent an older adult patient from becoming seriously ill or to prevent their sudden deterioration due to a condition and to ensure early discovery?”.

The first author of the research, who was a clinic doctor, conducted the interview on Zamami Island, and verbatim records of the interviews were created based on audio recordings made with an IC recorder.

### 2.4. Data Analysis

The interviews were audio-recorded with the consent of the participants. Based on these recordings, verbatim records were created and analyzed using the steps for coding and theorization (SCAT), an analytic method used in qualitative studies introduced by Otani in 2008 [18,22]. We used the SCAT method to analyze the content of the interviews. It consists of the following four steps:

Step 1: Identify focus words and phrases from within the interview texts.

Step 2: Identify words and phrases derived from the words and phrases chosen in Step 1.

Step 3: Identify concepts that explain the words and phrases in Step 2.

Step 4: Deduce themes and concepts, including the process of writing a storyline, and offer theories that interweave these themes and constructs.

Concepts were extracted from interview data and then categorized into themes. The main author conducted these four processes initially. Next, the coauthors who were well-experienced in the qualitative research, reviewed the coding for theoretical triangulation. When opinions regarding the coding differed, the authors discussed these conflicts and agreed on the final concepts and themes.

### 2.5. Ethical Considerations

This study was conducted following the “Ethical Guidelines for Medical and Health Research Involving Human Subjects” of the Ministry of Health, Labour and Welfare of Japan, such that the data of research participants were kept confidential. We obtained written consent from the participants after informing them of their right to refuse to participate in the study and that they could withdraw at any time. All personal data were anonymized and linked. Only the authors used the data. The paper data were stored in a locked cabinet, and all electronic data, including the link table created during linkable anonymization, were password-protected. If participants refused to participate or withdrew from the study, all their data, including personal data, were discarded immediately. This study was conducted with the approval of the Okinawa Prefectural Nanbu Medical Center & Children’s Medical Center Ethics Committee (Approval No.: 2018–52).

## 3. Results

As a result of the SCAT analysis, we extracted 13 concepts, which were categorized under four themes (Table 1). The following text shows quotes from the interviews in quotation marks (“).

### 3.1. Collection and Communication of Information between Residents

#### 3.1.1. Watching and Taking Care of Each Other Based on Close Relationships

The accessibility afforded to people residing on a small, remote island has allowed them to establish various relationships among residents, such as friends, relatives, neighbors, or coworkers. They showed concern for one another and watched out for each other according to their relationships.


*“We look out for each other, asking questions such as ‘How is that old man?’ We have concern for one another because we have lived in the same island.”*



*“There tends to be a lot of people who get sick without us becoming aware, and at times, the community residents tend to know more about one another. The community is small enough for each person to know each other well, and quite often, islanders notice changes in the islanders’ conditions faster than we (HCPs) do.”*


#### 3.1.2. Residents Consulting Local Leaders

In areas with neighborhood chairpersons, such as the head of the district, residents relay the information to these heads. Subsequently, these chairpersons consult HCPs based on the information from the islanders.


*“The head of districts have to distribute local works for people who have lost their jobs and need money. So, they naturally step in-to a private situation.”*



*“She worked as the head of a district for more than 10 years. Thus, lots of residents trust and rely on her, and they still tell her some essential information.”*


On remote islands in Japan, culturally, the head of the district is selected by citizens’ polls. Usually people are elected who have lived in that place for a long time and who are trusted by citizens. On this island, this applied, and they were in charge of financial support for islanders in each district. Local leaders can therefore easily obtain personal information about islanders, including health-related concerns, compared to general islanders.

#### 3.1.3. Presence of Potential Healthcare Human Resources

According to the data provided by the local government, Zamami Island conducted a care worker course in 1999. During that time, approximately 90 people obtained a careworker’s license. Despite these people having no record of working as home care workers on the island, the interview participants mentioned that the presence of these qualified individuals on the island encouraged residents to carefully observe the condition of older adult residents.


*“All of us took care worker courses together back in the day. Even if it was not used for work, it helped in providing caregiving for our parents. I think it is also useful that there are some people with nursing qualifications.”*


### 3.2. Communication of Information from Non-HCPs to HCPs

#### 3.2.1. Limitations of Communication between Non-HCPs

The residents took care of one another. However, they were wary about handling health-related information due to concerns regarding their privacy, and they avoided casually exchanging information with other residents.


*“The relationship (between islanders) may be too close. There might be a bit of hesitation since knowing too much about one another might affect the existing relationship.”*


#### 3.2.2. Ambiguous Boundaries between the Work and Private Lives of Professionals with a High Affinity to the Region

Each participant utilized their networks of information according to their type of occupation, the number of years of experience, length of residence on the island, presence or absence of relatives, and occupation of relatives and family members. All participants had a record of living on Zamami Island for at least a year. Thus, they were able to establish a network based on their work history and personal relationships, and they could easily identify and contact people to get certain information necessary for their work. This ambiguous boundary between the work and private lives of HCPs allows them to take a flexible approach to collecting information. The HCPs were well-integrated into the community, and residents could identify the HCPs among the islanders to provide them with information when required. The HCPs’ high affinity to the region also contributed to their flexibility in collecting information. They were able to pick out health-related information via word-of-mouth and, when necessary, actively collected information while ensuring the privacy and confidentiality of personal information.


*“I’ve lived here 20 years and have been working as a care worker for 15 of those years. Because of this, I know who to turn to when I need to ask something. You become familiar in 5 to 10 years, but not in 2 to 3 years.”*



*“I heard about Mr. A from the girl at the supermarket. I went to visit him at home after the girl said that he was acting unusual because he didn’t look at her and didn’t seem to be eating either.”*



*“A lot of islanders know that I take care of elderly, such as Mr. B and C. Thus, people told me wherever I was; not only at my office but also at the street or the post office, when their condition seemed to be bad.”*


#### 3.2.3. Preparation for the Collection of Information by HCPs

To facilitate the collection of information, in addition to the relationships that they naturally established with the residents, HCPs made daily efforts to establish and strengthen rapport with residents through their work and in their personal lives. Several interviewees mentioned the phrase “When I urgently need information, I collect information from the community by asking someone who seems to know about it.” This suggests that HCPs knew that the community would likely be knowledgeable about one another’s health status, and they collected this information not only passively but also actively and intentionally.


*“When a guy told me that the light of an old lady’s house continued turning on at midnight these days, I felt that the guy was taking care of and watching over the neighbor. I told him to let me know immediately if anything happened to the old lady.”*



*“We always exchange greetings, whether or not there is a problem. For example, there was an older man who cared deeply for his goat. He wasn’t very friendly with people, but he was open to showing his goat when I asked to see it. This sort of everyday connection might be important. This could help avoid rejection. Being acquainted with one another before any problem occurs is much better than meeting for the first time after the problem occurs.”*


#### 3.2.4. History of Public Health Nurses

Okinawa has previously established an original public health nursing system, which suits the environment of these remote islands. This was established after the end of World War II to improve public health status [23]. Public health nurses were stationed on each remote island and were colloquially known as “Ko-kan san” [24]. Although their purpose and role differed from those of modern-day public health nurses, the presence of a “Ko-kan san” in regions including remote islands helped streamline the work of public health nurses. Furthermore, elder islanders’ experiences with “Ko-kan san” helped in the smooth flow of information with these public health nurses.


*“’Ko-kan san (original public health nurses in Okinawa)’ used to go around changing bandages for bedsores and changing diapers. There were public health nurses even at a time when there were no caregiving insurance and no doctors on the island. The current situation might be a remnant of that. I think this is why islanders rely on modern-day public health nurses during times of trouble.”*


### 3.3. Sharing of Information between HCPs

#### 3.3.1. Conducting Regular Meetings

The interviewees in this study held interprofessional care conferences among themselves more than once a month [25]. They said that these regular meetings were an effective opportunity for exchanging and sharing information.


*“I tend to look through the minutes of the meetings even if I cannot go to the meetings themselves.”*


#### 3.3.2. The Connection between HCPs as Residents

HCPs were also residents of the island, and establishing relationships beyond that of work colleagues made the exchange of information easier. On remote islands, facilities related to administration, health care, medical care, and caregiving/welfare are often located close to each other, which makes it easier to visit workplaces and closely communicate outside of regular meetings.


*“We always share information. We always speak out if we notice something and speak to one another on the streets. If I am close by, I tend to stop by their workplaces when I am worried or even call them if the situation is urgent.”*


#### 3.3.3. Public Health Nurses as an Information Hub

HCPs were generally aware that they should “report to the public health nurse when an event or issue occurs,” as public health nurses served as the information hub for all older adult residents on the island.


*“The public health nurse is the go-to person when something happens. The information always goes through them.”*


#### 3.3.4. Ingenuity and Consideration in Sharing Information among HCPs

The information obtained by each person was handled carefully, considering to whom, when, and where it should be shared. During the recording of information, personal information and confidentiality were also taken into consideration.


*“We tend to share it once. There is a lot of information about older adults. If the person in charge is already handling the situation, we leave it as it is. I tend to make visits personally if I know the person well. We share information in that manner and start getting involved.”*



*“We don’t discuss every single piece of information together. We manage the information with the individual’s permission if we judge that it is necessary.”*


This showed that non-HCPs and HCPs share personal information not for private usage, but for professional usage with due consideration of privacy.

### 3.4. HCPs Taking Action to Initiate Their Approach

#### 3.4.1. Division of Roles through Various Meetings

Regular meetings are held to initiate an approach through interprofessional collaboration after collecting and sharing information through work and private conversations, as well as to assign roles.


*“The public health nurse calls for meetings and formulates care plans, and the nurses at the clinic administer medications. I think it is a good example of interprofessional collaboration.”*


#### 3.4.2. Efforts for Better Collaboration with Facilities Outside the Island

It appeared that the facilities on the island collaborated smoothly to some extent through regular meetings and in other situations as described above. However, if a patient needs to be hospitalized or requires long-term institutionalization, it would be necessary to send them to hospitals and facilities off the island. An interviewee suggested that further collaboration is necessary in sharing information between *facilities in those situations.*


*“The hospital staff in the mainland may not be aware of the situation on the island, such as the lack of resources. Sometimes, they send patients back to the island without enough preparation.”*



*“Care managers exchange information with the hospital outside the island when our patients need admission. This is done because it is difficult when they are sent back to the island without calling in advance even though we don’t have enough facilities. Thus, we call and write to the hospital frequently.”*


### 3.5. The Conceptual Diagram of This Study

The findings of this study led to the creation of the conceptual diagram shown in Figure 2. Both non-HCPs and HCPs are residents on a remote island. Islanders including non-HCPs take care of each other while HCPs gather information both through their work and through their personal lives. After sharing and summarizing the information between HCPs, they assign roles and initiate an approach. To preserve confidentiality, HCPs do not freely provide information about older adults to non-HCPs.

## 4. Discussion

This study shows that information regarding the health of older adults was collected by islanders (both HCPs and non-HCPs) through interactions that depend on their relationships. This information was shared carefully among HCPs with the purpose of supporting older adults. First, the proximity of residents to one another allows them to care for and exchange information with each other. Second, when residents notice changes in their neighbors, they relay this information to HCPs. This smooth transition of information is brought about due to their close relationships, which were established through both their work and their personal interactions. Third, HCPs share this information and work more when necessary. Finally, HCPs take action to support older adults by collaborating with institutions on and off the island. This research is significant not only because it shows seamless communication among HCPs and islanders, but also because there is no previous research clarifying the current situation of interprofessional collaboration in remote islands.

On this remote island, the collection and communication of information between residents were based on the connection between people, which serves as social capital. Social capital is a concept coined by Putnam that considers “personal connections’’ as one of the factors related to community health [26,27]. The connection between residents is strong due to the enclosed environment of the island [28]. For example, in such a small community, close relatives and friends often share the same workplace. This proximity allows to them to watch out for one another and be aware of one another’s health conditions. Kawachi cites factors such as trust between people, mutual norms, and mutual help as examples of social capital contributing to health [29]. The relationships and interactions among neighbors in rural areas, in particular, are said to be more reliable than those in urban areas [30], which is a trend that supported the responses provided in the interview in this study. Additionally, non-HCPs such as residents provide lay care by taking care of each other and observing one another’s health condition [31].

Moreover, depending on the connection between residents and HCPs, residents reported information about their fellow islander’s health conditions to the HCPs when necessary. This smooth transition of information is brought about by the close relationships that were established both spontaneously and through the efforts of HCPs. In this research, a resident observed that a certain neighbor did not turn on their light and reported this to the HCPs. Residents were also aware of the HCPs’ backgrounds such as their job, social network, and their place of residence. This is a function of the length of time the HCPs have lived in the same community and their professional skills that allowed them to lay the groundwork to establish good relationships with residents [32].

Furthermore, HCPs try to prepare for cases of emergencies, regardless of whether these are work-related. Thus, non-HCPs can flexibly and intentionally choose who to relay information to [33]. Our study found that the collection of patient’s information by HCPs out of office hours is important. The job of HCPs on the remote island overlaps with their private life, and there is not much distinction between their job and private life. Additionally, the border of each professional’s role in rural area is vague [10]. A previous study shows that insufficient manpower in a rural area leads to this vague boundary due to organizational citizenship behavior (OCB) [18]. OCB is defined as people behaving in a certain way that can improve organizational functions despite being unsure of whether they will be rewarded for it [34]. HCPs and residents share their roles and are willing do more than the role entails [18]. On a remote island where manpower is limited, OCB may be more effective than it is in other settings. The collaboration between non-HCPs and HCPs in other settings should also be studied.

With regard to sharing information between HCPs, connections between HCPs also contribute to the quick information collection and smooth flow, which allows them to support and maintain the islander’s health. A previous report has indicated the importance of open communication among multiple healthcare professionals in remote areas [32]. As remote areas tend to have a small number of professionals, this naturally breeds a sense of fellowship among people. Furthermore, their shared environment dissolves the barriers that naturally exist between different occupations, ultimately leading to the better facilitation of communication [32]. In this study, we found that in this small remote island, effective multi-professional collaboration was achieved in a similar manner through the enhancement of both formal and informal communication among healthcare professionals. Additionally, a previous study shows the blurring of boundaries between roles and shared responsibility across disciplines can motivate rural clinicians who often manage intense workloads [35]. Hence, in terms of physician’s work, these ambiguous job boundaries can decrease the work pressure on rural clinicians. Such ambiguous boundaries promote efficient communication both among HCPs and between HCPs and residents.

After analyzing the information they collected, HCPs took action by assigning the appropriate roles to each HCP. This study shows that HCPs struggled with collaborating with facilities outside of the island. For instance, a previous study mentioned that there is a gap in the understanding between rural and hospital nurses regarding the settings and information that they need [10]. In this study, this pattern occurred between the HCPs on the remote island and the hospital staff on the mainland. According to their interviews, HCPs on the remote island made efforts to try to establish a good relationship with the mainland HCPs. Further studies should investigate the relationships of HCPs and interprofessional collaboration not only in the island but also off the island.

This study has some limitations and there are pointers for future research. First, as the interviews were conducted by a clinic doctor working in the same community during the same period, interviewees may have been hesitant to express negative sentiments. For example, we might not inquire how HCPs respond to and overcome the workload stress increased by vague borders between their professional and private lives. Despite asking them to be as honest as possible before the interviews, we think it is necessary for future studies to arrange for more interviews, if possible.

What is more, in our situation, the 10 participants were almost all the HCPs on the island, except for care workers and the physician. It is possible that a small number affected the results. However, almost all HCPs in the island were included, and we concluded that we could conduct a triangulation.

Third, even though we ensured the privacy and confidentiality of the information that was collected from the residents, it is possible that we may not have received entirely accurate responses from the interviews. However, as the interviews were conducted with people who usually work together, we do not believe that the validity of the study’s findings was affected by this potential limitation.

Fourth, this study cannot be generalized to all islands because all our participants belong to one remote island. We plan to conduct similar research on other islands in the future to obtain a general overview of the communication among HCPs and islanders and in other remote islands. In addition, this study can be used as a basis for future research to investigate the distribution of social capital on all remote islands, as well as the workload of HCPs and how they cope on remote islands and in remote areas.

Finally, in order to improve the credibility of this research, we could adopt a longer duration to search for data and perform analysis.

## 5. Conclusions

Islanders take care of each other and are aware of each other’s health status. At the same time, HCPs also attempt to collect the health information of islanders. When necessary, HCPs on the remote island gain essential information about older adults with health problems from islanders not only through their work, but also through personal interactions with the islanders. Subsequently, HCPs approach older adults who need health care. The human connections on the remote island serve as social capital and enable flexibility in both gathering information and facilitating seamless communication among islanders who also represent informal resources that contribute support for older adults.

## Figures and Tables

**Figure 1 ijerph-18-09479-f001:**
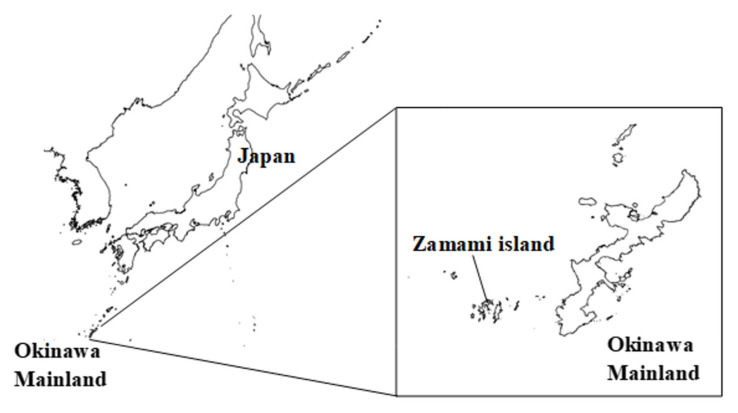
The location of Zamami island.

**Figure 2 ijerph-18-09479-f002:**
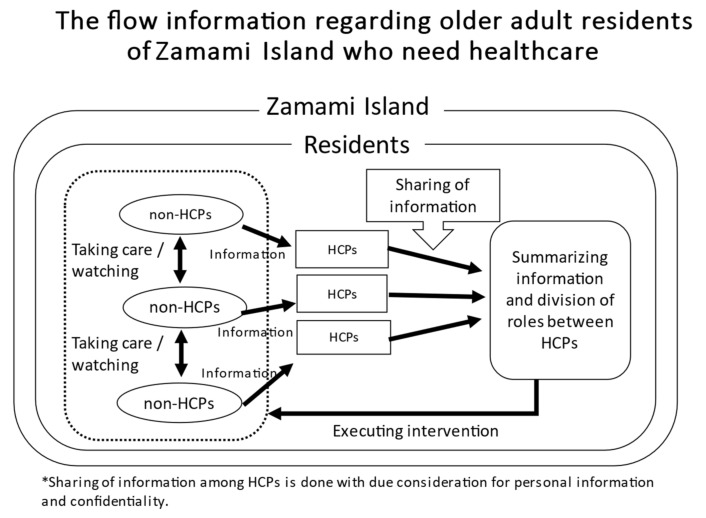
Conceptual framework for communication to support older adults in Zamami Island.

**Table 1 ijerph-18-09479-t001:** Results of the qualitative analysis: theme and concept.

Themes	Concepts
Collection and communication of information between residents	Watching and taking care of each other based on close relationships Residents consulting local leaders Presence of potential healthcare human resources
Communication of information from non-HCPs to HCPs	Limitations of communication between non-HCPs Ambiguous boundaries between the work and private lives of professionals with a high affinity to the region Preparation for the collection of information by HCPs History of public health nurses
Sharing of information between HCPs	Conducting regular meetings The connection between HCPs as residents Public health nurses as an information hub Ingenuity and consideration in sharing information among HCPs
HCPs taking action to initiate their approach	Division of roles through various meetings Efforts for better collaboration with facilities outside the island

## Data Availability

All relevant data sets in this study are described in the manuscript.

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
