# Peer review of "The Seamless Communication on a Rural Island in Japan: A Qualitative Study from the Perspective of Healthcare Professionals"

_ijerph, 2021, doi:10.3390/ijerph18189479_

Round 1

Reviewer 1 Report

This paper investigated a very important topic. Old people living in remote islands are often neglected due to a small population and insufficient healthcare professionals there. A study focusing on how to provide good care to this group of people is very important and may provide meaningful practical and theoretical implications. Despite the importance of the topic, I have several suggestions for the authors to further improve the quality of this manuscript.

Firstly, I think it is important to include relevant studies on this topic. Are there any studies focusing on healthcare in remote areas? Maybe you can cite more relevant studies in the introduction. In this way, you can better place your study among other studies.

Secondly, I think the information regarding the interviewee should be placed in the method section instead of the results section: moving lines 139-143 to section 2.2.

Thirdly, in the discussion, I think the authors could propose some future research directions based on the findings and limitations of this study.

Overall, I think it is a very good paper and I enjoy reading it very much.

Reviewer 2 Report

#Comments to authors

It is a great opportunity to review a manuscript, “The importance of seamless communication and information sharing in interprofessional collaboration among islanders and healthcare professionals on a rural island in Japan: A qualitative study from the perspective of Zamami islanders and healthcare professionals.” This is an interesting study and provides inputs into healthcare services. These are a few comments:   

  • The introduction:
  • It is fine but not very strong. It may be interesting to review some studies that were conducted on similar geographical situations, i.e., island and remote areas.
  • The method and material
  • The study design and methods were well structured and explained.
  • Results

The key information was clearly presented.

  • Discussion
  • L164: The people trusted the HCPs but we did not know how much the people trusted each other in terms of information sharing, particularly about health-related issues. Would it be a case that if one talks about another one’s issues with their peers and thus it becomes gossip?
  • L329-336: it is very interesting. It would be even more interesting to readers if the authors elaborate a bit more on how HCPs overcome the workload and private life.

Reviewer 3 Report

The manuscript does not meet publication criteria.

Round 2

Reviewer 3 Report

Many thanks to the authors for improving the manuscript.
I still do not find information about how the data from the recorded interviews is protected and stored, e.g. on digital media, on your university's repository, is it given an ID number, is it a backup, etc.
 On lines 288-289 and 315-316 as indicated by the authors, I do not find this information.
Please add this information to the methodology section.
